# Antimicrobial Resistance in Humans and Animals: Rapid Review of Psychological and Behavioral Determinants

**DOI:** 10.3390/antibiotics9060285

**Published:** 2020-05-27

**Authors:** Julie A. Chambers, Margaret Crumlish, David A. Comerford, Ronan E. O’Carroll

**Affiliations:** 1Department of Psychology, University of Stirling, Stirling FK9 4LA, UK; j.a.chambers@stir.ac.uk; 2Institute of Aquaculture, University of Stirling, Stirling FK9 4LA, UK; margaret.crumlish@stir.ac.uk; 3Department of Economics, University of Stirling, Stirling FK9 4LA, UK; david.comerford@stir.ac.uk

**Keywords:** antibiotic resistance, antimicrobial resistance, antibiotic stewardship, antimicrobial stewardship, human, animal, psychology, intervention, vaccination

## Abstract

A rapid review of current evidence examining psychological issues regarding the use of antibiotics and antimicrobials and resistance to these in both human and animal populations was conducted. Specific areas of interest were studies examining psychological determinants of AMR and interventions which attempt to change behavior with regard to AMR in the general population; animals; and fish, in particular. Although there is some evidence of the effectiveness of behavior change in general human populations, there is limited evidence in farmed animals, with a particular dearth in fish farming. We conclude there is an urgent need for more psychological research to identify major barriers and facilitators to change and evaluate the effectiveness of theory-based interventions aimed at reducing AM use in food production animals, including the promotion of alternatives to AMs, such as vaccination.

## 1. Introduction

Antimicrobial resistance (AMR) is an escalating risk to global health in both animals and humans, and progress to tackle its potential threat to life appears to be slow. Many bacteria have developed means of resistance to antimicrobials (AMs), which can spread to other bacteria, reducing the effectiveness of subsequent AM treatments in both humans and animals. Antibiotic (AB) and AM use in animals has important implications for AMR because of the potential of inter-species transmission of resistance. AM use in animals has been increasing at a concerning rate and so poses a considerable threat to AMR in both animals and humans. Accordingly, infectious disease experts have called for “strategies to minimise the risk of spreading antibiotic resistance among all populations” (i.e., humans and animals) [1] (p. 47). Although the mechanisms which lead to AMR are biological, the motivations behind current levels and methods of antimicrobial (AM) usage in both humans and food animals are determined by a wide range of factors including individual, psychological, social, cultural, political and economic forces. Many of these factors are likely to differ across human AB use and AB use in animal populations. 

Psychological factors play an important role in determining behavior. This literature review will add to existing evidence on associations between psychological factors and issues surrounding AM usage and AMR development, in members of the public as well as animal producers and owners, a little-studied area. Few studies have examined the role of psychological factors as drivers for AMR in humans, and even fewer have examined their consequence in AMR development within the animal food production sector [2]. Veterinarians’ and farmers’ decisions about using AMs in animals are just as likely to be governed by their knowledge, attitude, and beliefs as those making decisions about AB usage in humans. Thus, by including psychological determinants of behavior towards AMR in both humans and animals, this review will serve to highlight important discrepancies in understanding, and so help inform interventions which may be needed to change behavior.

The concept of AB or AM stewardship is concerned with the judicious use of AB/AMs in order to ensure their continued efficacy in both humans and animals. It is all-encompassing: contexts in which ABs are used including hospitals, general practitioner surgeries, dentists, the livestock industry (including fish), veterinary surgeries, and other groups such as prisons, nursing homes, and even racehorses and domestic pets. In food production not only veterinarians, but farmers themselves, are responsible for administering antibiotics; in humans, patients’ behavior (e.g., expecting or requesting ABs for viral infections) may place undue influence on health professionals’ prescribing behavior. It is therefore important to understand the role of non-health professionals in determining AB usage, and this forms the focus of this review.

### Search Strategy

The aim was to provide a rapid review of the research evidence examining psychological issues surrounding the understanding of the use (appropriate or otherwise) of ABs/AMs and attitudes to AMR in the general public and animal farmers/owners. A rapid review was chosen as it provides a knowledge synthesis in a timely manner by simplifying the systematic review process [3]. Appropriate AB or AM usage includes: not using ABs/AMs for diseases in which they are ineffective (such as viruses); completing a full course; not using leftover ABs from a previous infection, and not insisting that ABs/AMs are prescribed for all diseases. Studies that focused exclusively on health professionals and veterinarians were not included. This is not intended to be a detailed systematic review, but rather a broad overview of the currently-available evidence in order to inform future empirical studies. Both quantitative and qualitative research was included in this review, as, although the former is likely to provide evidence-based recommendations, the latter provides richer data and understanding, particularly where people’s attitudes and beliefs are concerned, and so can contribute significantly to the overall picture. 

Specific areas of interest were studies examining psychological determinants of AMR and interventions which attempt to change behavior with regard to AMR in the general population and all animals, including farmed fish. In particular we searched for theoretically-based studies, as these provide better scientific rigor, and these have been reported separately. Theory-based studies are important because they offer the potential to determine not only if behavior changes but also the mechanism of change, e.g., via potential mediating constructs such as perceived necessity, concerns, attitudes, intention, and cost. This can then inform interventions that can be targeted at modifying key constructs, e.g., reducing AB necessity and increasing concerns beliefs. A particular focus of this review is to inform our Vaccines against AMR in Aquaculture project (VAAC), which aims to deliver novel aquatic vaccines and test innovative machine vaccination to reduce antibiotic use in the Vietnamese catfish sector. As part of this project, Vietnamese catfish farmers’ attitudes towards antibiotic use, AMR, and vaccination will be examined. Knowledge, attitudes, and beliefs of farmers are likely to differ between Western and developing cultures, as government legislation will vary; therefore, it was important for us to pay particular attention to studies conducted in developing countries and in aquaculture.

Search terms for this rapid review included ‘antibiotic’, ‘antibiotic stewardship’, ‘antibiotic resistance’, ‘antimicrobial’, ‘antimicrobial stewardship’, ‘antimicrobial resistance’, ‘AMR’, ‘human’, ‘animal’, ‘farming’, ‘fish’, ‘aquaculture’, ‘psychology’, ‘psychological’, ‘psychosocial’, ‘attitudes’, ‘knowledge’, ‘beliefs’, ‘intervention’. Reference lists of relevant articles were also searched for additional research papers not identified during the main searches. All articles reviewed have been published since 2007.

## 2. Results of Review

### 2.1. Psychological Studies on AMR in People 

Studies included in this rapid review which involve the use of ABs/AMs in humans are summarized in Table 1; they are listed in the order they appear in the text.

There have been a number of recent systematic reviews and meta-analyses examining knowledge, attitude, or behavior towards antibiotic (AB) usage and/or AMR in the general population. Gualano et al. [4] conducted a review and meta-analysis of 26 cross-sectional, questionnaire-based studies conducted in the general population in Europe, North America, Asia, and Oceania. They found a lack of knowledge regarding the purpose of antibiotics with a third of participants being unaware that they were used to treat bacterial infections and over half not knowing they were ineffective against viruses. Although 60% stated they were aware of AMR, over a quarter did not know that AMR may be caused by inappropriate use of ABs. Almost half said they stopped using ABs when they felt better and did not complete the course. 

McCullough et al. [5] conducted a systematic review of 54 quantitative and qualitative studies conducted in Europe, Asia, and North America which examined the public’s knowledge and beliefs about antibiotic resistance (ABR). Although they found that typically more than half of the participants said they were aware of ABR, the vast majority of these believed it referred to changes in the human body. More than half believed that overuse of ABs and not completing a course caused resistance and that reducing the use of ABs would be helpful in reducing resistance. However, the qualitative studies revealed that people believed their own risk from ABR was low and that ABR was due to the actions of others. They also stated that strategies to reduce AM usage needed to be focused on clinicians.

Two systematic reviews examined parents’ attitudes, knowledge, and behavior towards AB use in their children in qualitative and quantitative research in the UK, US, Europe, Asia, the Middle East, and South America [6,7]. Most parents sought early medical aid for their child’s illness; however poor parental knowledge of ABs (such as believing them to be a ‘wonder drug’ for all ills) and concerns for the seriousness of the child’s illness led to greater AB usage. Parents also had expectations that clinicians would prescribe ABs, often based on previous experience, and again many stopped the use of ABs when the child seemed better, rather than complete the course. Less developed countries and lower socio-economic groups had poorer knowledge of ABs and ABR, probably due to lower education levels, but good knowledge did not necessarily lead to more appropriate usage, such as not using them for colds and viruses and not using leftover ABs. In fact more educated parents were more likely to self-prescribe with over-the-counter products, perhaps because they felt confident in their knowledge of AB usage. Improved reassurance from clinicians on the appropriate prescribing of ABs, as well as risks and side-effects, could lower parents’ expectations and use of ABs, but this was not always given. Although awareness of ABR was often high, few parents believed that giving ABs to their own children contributed to the problem—a similar finding to studies with the general public. A later study with US parents confirmed the results regarding awareness of issues surrounding AB usage and needing clinician reassurance but also found that parents did not have concerns about ABR or AB treatment failure [8]. 

A recent qualitative study examined the drivers of AMR in pet owners (n = 23) and veterinarians (n = 16) across the UK [9]. They found that understanding of AMR in general amongst pet owners was low and understanding of the role of pets in driving AMR was almost non-existent. Both pet owners and veterinarians tended to believe it was the other party who was responsible for the inappropriate use of ABs. 

Government policies regarding AB and AM usage differ between countries, and this may contribute towards inappropriate AB usage on a global scale. This may be a particular issue in developing countries: with lower overall levels of education amongst the general public, and potentially less access to medical care, it might be expected that poorer AB/AM awareness and more inappropriate use of ABs would be more evident in such countries. However, there is less research conducted in these countries on which to base evidence. A few of the studies included in the above reviews with the general public were conducted in Asian countries, including Malaysia, China, Indonesia, and South Korea [4,5]. Findings were not dissimilar to Western countries, including poor knowledge of the effectiveness of AMs against viruses and unawareness of AMR, as well as inappropriate AM usage; these findings were replicated in a recent large (n = 3390) national survey in Japan [10]. A small (n = 20) qualitative study in urban and rural India [11] examined social determinants of AB usage and found poor understanding of both ABs and the implications of misuse. In particular participants in rural areas, with limited access to doctors, were more likely to self-prescribe and stop taking ABs early. However, all participants were more likely to seek a doctor’s advice for their children than themselves.

#### 2.1.1. Theory of Planned Behavior (TPB)

The TPB [12] is an established and widely-used theoretical model that underpins robust research into psychological determinants of behavior. The TPB posits that behavioral intention is strongly related to carrying out a specific behavior, and is influenced by attitudes, social norms, and perceived behavioral control. Briefly, a behavior, such as reducing AB usage, is more likely to be performed if it is perceived to be beneficial, supported by one’s peer group and likely to be achievable, for example if there were satisfactory alternatives. The TPB proposes that individuals make logical, reasoned decisions to engage in specific behaviors, but it has been criticized for not including the role of emotion/affect.

Although there are a few studies examining the TPB in healthcare practitioners [2], with inconsistent results, no studies looking at the TPB in relation to AMR in the general public were found. However, one study did examine TPB constructs in relation to intended AB usage: Byrne et al. [13] recently developed a TPB-based measure aimed at assessing the drivers of AB usage in the community, which they believe should encourage further relevant research in this area. In this preliminary study, they observed that social norms, perceived behavioral control, and attitudes and beliefs moderated by the covariate knowledge, were all significant predictors of self-reported AB usage. People’s attitudes and social norms affected their intention to use ABs over and above accurate knowledge, suggesting interventions aimed at increasing knowledge of appropriate AB usage are unlikely to be effective without also targeting motivations and social influences. 

#### 2.1.2. Interventions

There have been a number of reviews of educational interventions to improve knowledge, attitudes, and beliefs, as well as behavior towards AB usage and AMR. 

A recent Public Health England report examined current behavior change research and its application to reducing AB usage [14]. Although the authors found that many public-targeted social marketing campaigns aimed at improving understanding and better usage of ABs have been conducted over the last 20 years, there was no strong evidence that such interventions are effective. There was limited evidence that increasing vaccination usage was linked to lower AB use in the US. In contrast, interventions targeting clinicians did tend to lead to more appropriate AM usage. The authors also found that public understanding of AMR was patchy, with confusion over the difference between bacteria and viruses and poor understanding of the meaning of AMR. They recommended that approaches which work via automatic, rather than reflective, psychological processes, such as changes to the environment and existing systems, may be most cost-effective in changing behavior and improving AMR stewardship (managing the appropriate use of AMs) in both clinicians and the general public.

A National Institute of Health and Care Excellence (NICE) review found that educational interventions led to increased knowledge about the appropriate use of AMs (although this remained patchy) but not to better knowledge of AMR [15] (King et al., 2015). Again, direct contact education programs were more successful than mass media interventions, which they attributed in part to the fact that many people remained unaware of the mass media campaigns. They found inconsistent evidence that education aimed at improving hand hygiene had any impact, but there was some evidence that targeted interventions could improve knowledge and behavior with regard to food handling and safety. This review did not appraise interventions which examined changes to AB prescribing rates. 

A recent systematic review of US educational studies aimed at increasing public awareness of AMR [16] found that educational interventions (e.g., via posters and handouts) did result in improvements in knowledge, attitudes or beliefs, but there was only clear evidence of decreases in AB prescribing if clinician education (including decision-making tools and prescribing guidelines) was also delivered. The risks of AB over-usage were not covered in all of the reviewed studies, most of the studies were conducted with parents, and any follow-up was short-term, so it was not known if observed changes in prescribing would persist over the longer-term. They concluded that, while simple education materials may be effective at changing knowledge, attitudes, and beliefs, a multifaceted approach, involving both patients and clinicians, may be required to reduce AB usage. A review of reviews for NICE [17] also concluded that whilst multi-component interventions targeted at the public increased awareness of AMR, only interventions involving both the public and clinicians seemed effective at changing AB usage. 

Price et al. [18] conducted a systematic review of the effectiveness of interventions to improve the public’s AMR awareness and use of AMs. They reviewed 20 studies that excluded health professionals, 19 of which were conducted in high-income countries (US, UK, Europe, and Oceania). Targeted groups included schoolchildren, parents, and the general public. Most studies found an increase in knowledge, attitudes, or improved behavior with regard to AMs (e.g., not taking an AB for colds, using hand sanitizers or being vaccinated against influenza). Educational interventions with schoolchildren and parents were the most effective whilst evidence of the impact of mass media interventions in the general public was less clear. Again long-term outcomes were not examined. Although only four of the studies reported any explicit theoretical basis (two social marketing theory; two ‘precede-proceed’ model), intervention descriptions revealed that all included at least one behavioral change technique [19]. The most common of these focused on educating participants about the consequences of behavior and using a trusted source of information to instruct participants on how to perform AMR-related behaviors. 

#### 2.1.3. Affect

People’s behaviors are not just directed by rational thought processes (such as knowing that ABs are not effective against viruses) but also governed by emotional responses (such as worrying unduly about a child’s health). The authors of the Price et al. [18] review also conducted an innovative analysis of the use of affect (i.e., emotion) within the visual content in the intervention materials of the reviewed AMR studies [20]. They found that emotion or affect was tacitly included in most of the studies, but was not formally theorized or evaluated. The authors described three affective themes: monsters, bugs (which included threat, disgust, and fear) and superheroes (e.g., ‘Andybiotic’ protector and needing protection); responsible stewardship, threat and misuse/use of ABs (which included guilt, (moral) disgust and social/moral disapproval); and use of child figures (moral/generational responsibility, parental caring/protection of the child). They found some association with the outcomes of interventions supporting the belief that warnings in images (e.g., a photo of a young, healthy child alongside the caption ‘Will he reach 20? Maybe not if antibiotics become ineffective. Use antibiotics wisely’) are likely to be more effective than text alone. The authors concluded that affect could be an important tool in eliciting behavior change with regard to AM usage. Although a small study, with a number of limitations including using previously untested methods of analysis, it raises the idea that affect should be explored in future interventions. 

#### 2.1.4. Implementation Intentions (IIs)

IIs provide a framework for developing interventions which address habitual behaviors and so can be effective at changing usual responses, such as always seeking a GP prescription for ABs for colds. IIs involve a person planning, in advance, when, where and how they will implement a particular behavior [21]. Chaintarli et al. [22] examined the effectiveness of a pledge system for clinicians and members of the public who had signed up to the Antibiotic Guardianship initiative in the UK. This initiative, launched by Public Health England in 2014, aimed to get people to register as Antibiotic Guardians on the website (www.antibioticguardian.com) and choose a pledge. Pledges were based on an Implementation Intentions (II) approach which aims to get people to decide, in advance when, where and how they will act in a particular situation using an If-then approach. For example one of the quoted public Antibiotic Guardian pledges states: ‘For infections that our bodies are good at fighting off on their own, like coughs, colds, sore throats and flu, I pledge to talk to my pharmacist about how to treat the symptoms first rather than going to the GP’. The authors found that both self-reported knowledge and a sense of personal responsibility increased and they concluded that online pledge systems could be effective in changing behaviors with regard to AMR. However, the sample was biased towards people with previous AMR awareness, 70% of respondents were females aged 44–54 and response to the survey required online access, so it’s not clear how generalizable it would be to other groups. Pinder et al. [14] also recommended that using online pledges in parents of children may be a useful tool in reducing AM usage, although this was not specifically linked to IIs. 

#### 2.1.5. Social Capital Theory (SCT)

Rönnerstrand and Andersson Sundell [23] examined the impact of trust and reciprocity on people’s willingness to postpone using antibiotics, using a social capital framework. SCT [24] is concerned with the norms that influence people’s behavior when there are dilemmas between self-interest and cooperation for the greater good as may be demonstrated in deciding to take ABs for short-term personal gain whilst considering the long-term effects of AMR for all. They used hypothetical scenarios where doctors prescribed ABs but asked patients to delay taking them, and found that participants with greater levels of generalized trust were prepared to delay longer, and that the length of time they were told others would delay (reciprocity) was also strongly related to their own reported length of delay in taking ABs. 

#### 2.1.6. Summary of Psychological Studies and AMR in Humans

Whilst many of the studies which have examined for knowledge, attitudes and behavior towards AM usage in the general public have recommended better information campaigns, there is mixed evidence regarding the effectiveness of mass media interventions [14]. Direct contact education was shown to be more effective than mass media campaigns. Interventions which targeted clinicians alongside the public had the most success in changing AB usage, and so changing public awareness, attitudes and behavior may not be possible in isolation. Further, most public-targeted campaigns are not based on behavior change theory [25]. 

Nonetheless, Pinder et al. [14] identified many areas for developing interventions targeting both awareness and behavior; they recommended that the public need a clearer understanding of appropriate AM usage before a social norm of ‘ABs as a last resort’ could be established. The study based on Social Capital Theory also supports the assertion that a collective identity towards AM usage could be used to achieve changes at a societal level [23] (Rönnerstrand & Andersson Sundell, 2015). The studies targeting parents suggested that their view of the necessity of AB treatment for their children was more important than concerns about the wider risk of AMR, but that this could be mitigated by clinician reassurance. Beyond this, there is a dearth of studies which are underpinned by psychological theories, and therefore much scope for new interventions, including targeting implementation intentions, social norms and affect. 

### 2.2. Psychological Studies on AMR in Food-Producing Animals

Studies included in this rapid review which involve the use of ABs/AMs in food-producing animals are summarized in Table 2; they are listed in the order they appear in the text.

There is currently a dearth of studies examining psychological determinants of behavior in relation to AMR in animals [2]. A rapid evidence assessment (REA) by DEFRA [26] on the use of AMs in animals observed that most research into farmer and veterinary behaviors regarding AM use has been conducted by natural or animal scientists, and not psychologists, resulting in clear knowledge gaps regarding how veterinarians’ and farmers’ beliefs and attitudes may influence current practice. Hockenhull et al. reviewed research relating to attitudes, beliefs, and external influences on antimicrobial use in land-based animals [27]. In summary, they found that drivers and barriers to AM reduction were related to practical factors, such as the cost and health of animals, rather than any beliefs about the risks of AMR. Whilst most veterinarians and many farmers in the study believed reducing AM use would be a good thing, the majority underplayed the threat of AMR to human health, often citing insufficient evidence. In some cases, despite antibiotics being originally prescribed by veterinarians, many farmers did not seek their veterinarian’s advice when deciding whether or when to administer treatment. Farmers also expressed concerns that reducing the usage of AMs would lead to unhealthy animals or lower production levels [27].

Most of the studies which touch on psychological issues, including attitudes, barriers, and motivations regarding AM use, have been conducted in Western cultures and on cattle, sheep, and pig farms. Coyne et al. [28] conducted focus groups with pig veterinarians and farmers and identified a number of themes relating to drivers and motivations of AM use. Whilst veterinarians appeared to be influenced by external pressures including clients’ and the public’s views as well as legislation, farmers’ concerns were more related to the production and management of their farm. Gibbons et al. [29] also found that veterinarians were susceptible to external non-clinical pressures especially farmers’ expectations. 

In a later mixed-methods study with UK pig farmers [30], Coyne et al. concluded that influences were complex, but that reduced AM use was associated with good management practices, low stocking densities and good animal health. The high cost of AMs could be a driver to reduce usage for farmers, but this was offset by improvements in pig health. Vaccination was also viewed as a feasible alternative to AM use by the majority of farmers, but few quoted concerns about AMR as a driver to reducing AM use [30].

Another recent UK qualitative study, conducted in the cattle, sheep and pig sectors [31], assessed veterinarians’ and farmers’ attitudes to their role in AMR resistance and stewardship. AM stewardship can be considered as the means of looking after the use of AMs in such a way as to preserve their efficacy in both human and animal contexts. In the Golding study, both veterinarians and farmers showed a good understanding of stewardship issues but their treatment behavior did not always tally with these beliefs. Fear of negative health outcomes led to both veterinarians’ and farmers’ inappropriate usage of AMs. Both veterinarians and farmers also reported that poor practice in others undermined their own efforts at stewardship. The authors concluded that tackling AMR requires intervention at the individual, group, and societal level, and that a social identity approach could address the issue of ‘other-blaming’. 

In the US, Friedman et al. [32] conducted four focus groups to investigate the sources of information and knowledge about antibiotic resistance in 22 dairy farmers. Although 40% of farmers were familiar with AMR, the vast majority (86%) were not concerned that the overuse of antibiotics in animals could lead to resistance amongst farmworkers. They also stated that the lack of finances and time were barriers to optimal use of AMs. The authors also found that advice from veterinarians and other farmers seemed more important than scientific reports when farmers were making decisions about using AMs, and they concluded that education (e.g., via posters, flowcharts, videos, seminars) on the appropriate use of antibiotics is required. Another qualitative US study examining drivers for AM usage in beef cattle producers also found that practical considerations, including economic factors, the influence of veterinarians and peers, and animal welfare, and not concerns over AMR, were instrumental in AM usage decisions [33]. Vaccination and good management practices were seen as valid alternatives to AM use. 

Another recent study examined associations between Dutch farmers’ beliefs and actual AM usage [34]. The authors used a questionnaire to examine motivating and enabling factors regarding AM usage in 457 dairy, veal, and pig farmers. The analysis suggested four psychological factors: ‘referent beliefs’ (external non-vet sources of advice), ‘perceived risk’ (AMR risk perception), ‘knowledge’ (of infection routes and effectiveness of AMs), and ‘undesired attitude to regulations’ (disregard for AMR regulations). Across all farming sectors, better knowledge was significantly linked to lower AM usage, and there was a tendency for a relationship between lower AM use with both higher perceived risk and a lower ‘undesired attitudes to regulations’ score. The authors concluded that, in order to engage in better practices, and hence reduce AM usage, farmers needed educational programs and/or increased support by veterinarians, via a collaborative, rather than paternalistic, communicative style, to help them make positive decisions about AM usage. In a cross-sectional survey across four European countries (Belgium, France, Germany, and Sweden), Visschers et al. [35] also found that pig farmers’ perceived risk of AM usage was related to the actual usage of AMs. In this study, farmers perceived many benefits of AM usage but relatively few risks. 

#### 2.2.1. Theory of Planned Behavior (TPB)

A few studies have examined the TPB or social norms in relation to AM usage. Jones et al. [36] examined attitudes and decision-making via questionnaire in UK dairy farmers (n = 71) and found only a weak association between past behavior and intention to use AMs, but that intention was strongly predicted by social norms, including perceived approval by veterinarians and other farmers for the use of AMs in specific instances. More commercially-aware farmers were more likely to intend to reduce their use, again suggesting that the cost of administering AMs could be a possible driver for reducing AM use. 

McIntosh and colleagues have conducted a number of studies in the US cattle feedlot industry, which examine TPB concepts including social norms and beliefs (e.g., Reference [37]). Overall, they found a strong influence of social norms (including veterinarians, consumers, pharmaceutical companies, regulatory bodies) in decision-making regarding AM use, but that these influences differed depending on the animal’s health (e.g., acute vs. chronically sick) i.e., who was viewed as important in giving advice would depend on the situation. Nonetheless, they conclude that in order to reduce AM use it will be necessary to change the beliefs of significant others. In an earlier study, the same group found that subjective norms and a sense of moral obligation affected both attitudes toward AM use and veterinarians’ recommendations as to AM usage in feedlot cattle [38]. 

Another study examined social factors which influence farmers’ decisions (n = 38) about how long to administer antibiotics in the treatment of mastitis in dairy cows in the Netherlands and in Germany [39]. They found a strong effect of the views of other farmers in influencing length of treatment, with extended treatment being associated with the social norm of being ‘a good farmer’. They concluded that this influence hindered appropriate antibiotic use. 

#### 2.2.2. Research in South-Eastern Asia

A few recent studies have considered attitudes towards AM usage in countries in Southeastern Asia. In many Asian countries, antibiotics can be used in animals without needing to be prescribed by veterinarians. In contrast to Western farmers, knowledge of AMR risk in Asian farmers appears poor. A questionnaire study of Malaysian ruminant farmers (n = 84) found little awareness of AMR and its impact on animals and public health [40] (Sadiq et al., 2018). Farmers, especially those with smaller farms, believed that all sick animals need antibiotics, that AMs do not have any side-effects, were indifferent to only using AMs when prescribed by a vet, and stored AMs on their farm for later use. 

A structured survey examining usage, knowledge, and attitudes of farmers in 91 small-scale pig farms in Cambodia found evidence of widespread inappropriate AM usage [41] (Ström et al., 2018). Nearly half of the farmers had not heard of AMR, and the farmers relied on their own judgement regarding antibiotic treatment. Nearly 60% thought they had not received enough information on the use of AMs in animals. The authors concluded that the low level of awareness of AM usage and risks of AMR in these farmers was of great concern. 

Finally, Coyne et al. [42] recently reported on AM usage in food animals (including pigs, chickens and fish) in three Southeastern Asian countries (Indonesia, Thailand, and Vietnam), though they did not specifically examine farmers’ attitudes to AM usage. As with many of the studies in Western cultures, they found that drivers of higher AM usage were profitability and disease and mortality prevention. The cost of AMs was low compared with other production costs meaning farmers thought there was an economic advantage to their use. 

#### 2.2.3. Interventions

To our knowledge, no studies aimed at changing animal farmers’ beliefs or attitudes towards AM usage and/or AMR have been published.

#### 2.2.4. Summary of Psychological Studies and AMR in Food Producing Animals

In Western cultures, where antibiotics are prescribed by veterinarians—though decisions regarding administration are often down to farmers—most veterinarians and farmers tend to be aware of AM issues and believe that reducing AM use is a good thing. However, many studies found both farmers and veterinarians do not support or are skeptical about a link between the use of AM in food-producing animals and AMR in humans—for example in the UK [28,31] and the US [32]—which could be a major barrier to reducing AM use. In contrast, although few studies are currently published, knowledge or awareness of AMR in Asian farmers appears to be very poor. Government legislation regarding the use of antibiotics in animals varies between countries and cultures, and therefore further research is needed to understand how these differences may affect a farmer’s knowledge, attitudes, and beliefs. 

Social norms are a key influence on AM decision-making in Western farmers and veterinarians, and it has been suggested that addressing this could be key to reducing AM usage. In addition, some farmers believe AMs are necessary for maintaining animal health and have concerns that reducing their use would lead to poorer animal health and lower production levels.

In Asian countries, poor awareness of AMR risk coupled with the low cost of AMs means that choosing to treat animals with AMs is a relatively easy option. It has been suggested that increasing the cost of AMs in these countries as well as increasing knowledge of AMR may aid in reducing AM usage.

### 2.3. Psychological Studies on AMR in Aquaculture

A literature search did not reveal any studies examining the psychological determinants of fish farmers’ attitudes or beliefs towards AMR. One study, which looked at farmers’ knowledge and opinions on antibiotic use in freshwater Vietnamese fish and shrimp farms (n = 94: 63 fish farms and 31 shrimp farms) who produce for the domestic market, concluded that farmers had poor knowledge of the purpose of using antibiotics on their farms [43]. Almost half (45%) of farmers interviewed did not believe antibiotics were effective in treating disease, but nonetheless most (72%) stated they used antibiotics on a regular basis either to treat or prevent disease. Those who did not use antibiotics reported their fish or shrimp were healthy so there was no need [43]. Only 16% said they were aware of current regulations regarding antibiotic use and none kept a logbook of their use of antibiotics. Farmers sought advice on using antibiotics mainly from drug manufacturers and sellers and not from more impartial sources such as veterinarians. Reasons given for both using and not using antibiotics on their farms were linked more to economic reasons than because farmers felt a need to comply with regulations or because of any expressed concerns about AMR. However, in this study, participants were not directly asked about antimicrobial resistance. 

#### Interventions

No interventions aimed at changing farmers’ attitudes to antibiotic use in fish were found anywhere.

### 2.4. Key Findings

Awareness of appropriate AB/AM usage (including ineffectiveness against viruses) and the causes of ABR/AMR is often poor in the general population, and this is more evident in lower-income and less educated populationsAwareness of AMR in farmers is also often poor, especially in developing countries; however, good knowledge does not translate into reduced AM usageThe general public’s expectations of being prescribed ABs for common viruses persistFarmers are more influenced by social norms, including other farmers’ opinions, than advice from veterinarians when deciding whether to use AMs in animals. Practical considerations like economic factors and the animal’s health are also more determinant of farmers’ usage of ABs than consideration of ABRPeople attribute the rise in ABR/AMR to other people’s behavior (including clinicians) and do not see it as a problem for themselves or their children; even those with good knowledge of ABR believe that overuse in individuals raises the personal risk for those individuals but do not believe it is a risk for other people who restrict their own AB usageFarmers do not associate over-usage of AMs in animals with AMR in humans, including farmworkers; vaccination is seen as a viable alternative by someThere is inconclusive evidence that mass media campaigns aimed at the general public are effective; educational interventions aimed at parents and/or children may have merit in changing AB behaviorsThere is a dearth of evidence on changing psychological determinants of behavior with regard to AMR in farm animals

## 3. Discussion

To date, there has been limited research conducted with regard to psychological issues surrounding appropriate AM usage or AM stewardship. There is some published research regarding awareness, concern, and behavior around AMR usage in the general public, but much less examining changing behavior regarding the consumption of AMs in food-producing animals, suggesting there is a relative neglect of changing the risk posed to human health via AMR-induced by farming practices. 

A key contribution of the current research is to draw together two distinct bodies of literature that each contribute to our understanding of psychological determinants of AB use, and hence, pathways to AMR. By doing so, we hope to highlight blind spots in current policy and practice (e.g., how AMR might enter via the backdoor of animal husbandry) and to help identify how research aimed at changing the behavior of food-producing farmers is needed to prevent AB misuse.

Regarding AB usage in both humans and animals, there is a clear need for better education to improve awareness as well as interventions aimed at changing behavior. These may be better achieved via using techniques such as IIs (if-then planning) to change usual behaviors, such as requesting ABs for colds or flu or using AMs in animal feed, and/or developing a new social norm that ABs should be used as a last resort.

This review examined the psychological determinants of AMR in the general public and animal producers and did not specifically look at studies or interventions which might consider attitudes towards alternatives to antibiotics. Nonetheless, many of the reported reviews concluded that inappropriate or excessive antibiotic use tended to increase when people did not believe there were effective alternatives. This has been shown to be an issue in lower socioeconomic countries where, for example, hygiene was problematic and there was an increased risk of bacterial infection, prompting more or less preventive use of second-line antibiotics [44]. 

Effective vaccination can also reduce the need for antibiotics. A recent review by Brewer et al. [45] provided an important overview of the role that psychological science can play in increasing vaccination uptake in humans. They reviewed three main possible areas for intervention via targeting: (a) thoughts and feelings; (b) social processes; and (c) facilitating vaccination directly (e.g., using prompts, reminders, and reducing barriers). They concluded that the strongest current evidence suggests that (c) facilitating vaccination directly is most effective at changing vaccination behavior.

## 4. Conclusions

The review highlights that there is a particular dearth of evidence to guide interventions to try and change food farmers’ attitudes, beliefs, and behaviors to AM use. We urgently need psychologically-informed studies to: (a) identify the major barriers and facilitators to change, and (b) evaluate the effectiveness of theory-based interventions aimed at reducing AM use in animal and fish farming, in particular, including the promotion of alternatives to AMs such as vaccination.

## Figures and Tables

**Table 1 antibiotics-09-00285-t001:** Reviewed studies of psychological determinants of antibiotic (AB)/antimicrobial (AM) usage and AB/AM resistance (ABR)/(AMR) in humans.

Authors, Year, Location	Study Type	Sample	Behaviors of Interest	Key Findings
Gualano et al., 2014 [4]; Europe, North America, Asia and Oceania	Review/Meta-analysis	26 questionnaire-based studies (16 interview, 7 telephone, 3 internet-based), 25 < n < 10,780 participants	Knowledge of AB, AMR and appropriate use of ABs in general population	Lack of knowledge of purpose of ABs Poor knowledge of cause of AMR Inappropriate usage of ABs, e.g., not completing course
McCullough et al., 2016 [5]; Europe, Asia and North America	Systematic Review	54 studies: 41 quantitative, 10 qualitative and 3 mixed-methods, n = 55,225 participants	Knowledge and beliefs about antibiotic resistance (ABR)	Poor understanding of ABR Some understanding of causes of ABR Belief that own risk was low Belief that others, including clinicians, responsible for ABR
Bosley et al., 2017 [6]; UK, US, Europe, Asia, the Middle East and South America	Systematic Review	20 studies: 16 quantitative (survey), 3 qualitative (focus groups/interviews) and one mixed methods (survey/focus group); n = 15,899 participants	Parents’ attitudes, knowledge and behavior	Parents’ knowledge of ABs was poor, and behavior was influenced by previous experience Concerns for child and seriousness of illness lead to greater AB usage Parents had expectations that clinicians would prescribe ABs
Cantarero-Arévalo et al., 2017 [7]; UK, US, Europe, Asia, the Middle East and South America	Systematic Review	43 studies: 31 qualitative (survey-based; 60 < n < 1000 participants), 12 qualitative (focus groups/interviews; average n = 36 participants)	Parents’ attitudes, knowledge and behavior for children with URTIs	Less developed countries/lower socio-economic groups had poorer knowledge of ABs and ABR Good AB/ABR knowledge did not lead to more appropriate usage Parents did not believe giving ABs to their own children contributed to the problem of ABR
Szymczak et al., 2018 [8]; US	Mixed methods: open and closed questions	Face to face interviews with n = 109 parents	Parent’s perceptions of ABs use in children with acute RTIs	Parents understood overuse of ABs was an issue Parents were not concerned about ABR or failure of ABs in own children
Smith et al., 2018 [9]; UK	Qualitative	Face to face interviews with pet owners (n = 23) and veterinarians (n = 16)	AMR behaviors and interactions between pet owners and veterinarians	Understanding of AMR amongst pet owners low Understanding of the role of pets in driving AMR almost non-existent Pet owners and veterinarians believed the other party responsible for inappropriate AB use
Kamata et al., 2018 [10]; Japan	Nation-wide questionnaire survey	Online survey; n = 3390 participants aged 20–69, not medical professionals	Knowledge and perception of AMR	Poor knowledge of the effectiveness of AMs against viruses Desire to have ABs prescribed for common cold Some adjust doses themselves
Barker et al., 2017 [11]; India	Qualitative study	Face to face interviews with community members; n = 20 participants	Social determinants of AB use	Poor understanding of ABs and the implications of misuse Participants with limited access to doctors were more likely to self-prescribe and stop taking ABs early
Byrne et al., 2013 [13]; Australia	Questionnaire-based survey	Paper and online survey; n = 373 adult participants	Theory of Planned Behavior (social norms, attitudes, perceived behavioral control) to assess drivers of AB usage	Social norms, perceived behavioral control, and positive attitudes and beliefs were all significant predictors of self-reported AB usage People’s attitudes and social norms affected their intention to use ABs over and above accurate knowledge
Pinder et al., 2015 [14]; countries not specified; low-income settings excluded	Literature review and behavioral analysis	54 intervention studies with primary care, hospital and community patients, no further details given in report	Understanding of behaviors relating to AB usage and AB stewardship including role of vaccination	No evidence for effectiveness of public-targeted social marketing campaigns Public understanding of AMR patchy, with confusion over the difference between bacteria and viruses Poor understanding of the meaning of AMR
King et al., 2015 [15]; US, UK, Australia, New Zealand, Europe	Systematic review	60 intervention studies with general public; 26 < n < 2006 participants	Behavior re: appropriate use of ABs and reducing ABR	Education increases knowledge of AM usage but not AMR Direct contact interventions more effective than mass media campaigns
Burstein et al., 2019 [16]; US	Systematic review	34 intervention studies, most educational; 19 targeting public; participant numbers not reported	Knowledge, attitudes and beliefs	Simple education interventions (posters/handouts) improved knowledge, attitudes or beliefs decreases in AB prescribing linked to clinician, not public, education
Haynes & McLeod, 2015 [17]; locations not specified	Review of reviews	9 review papers od educational intervention studies most educational; no participant information given	Knowledge and behavior	Multi-component interventions targeted at the public increased awareness of AMR but only interventions involving both the public and clinicians changed AB usage
Price et al., 2018 [18]; US, UK, Europe and Oceania (Further reporting in and McParland et al., 2018 [19] and Langdridge et al., 2019 [20])	Systematic review	20 intervention studies; 7 mass media, 6 printed material, 7 educational; 10 < n < 6217 participants	Knowledge, attitudes or behavior (e.g., not taking an AB for colds, or being vaccinated against influenza) with regard to AMs	Some evidence that educational interventions with schoolchildren and parents effective Impact of mass media interventions in the general public unclear Although not explicit, many of the reviewed studies included a behavioral change technique Images based on emotion may be more effective than text messages
Chaintarli et al., 2016 [22]; UK	Pre-post online survey (following campaign)	Adults signed up to Antibiotic Guardianship online pledge system; n = 2478 participants	Knowledge and behavior; Implementation intentions	Increase in self-reported knowledge Increase in a sense of personal responsibility towards AMR
Rönnerstrand et al., 2016 [23]; Sweden	Hypothetical scenarios in experimental study	Adults, n = 981 participants, mean age 51 years	Social Capital Theory; willingness to postpone AB treatment	Participants with greater levels of generalized trust would delay longer in taking ABs Length of time others would delay taking ABs (reciprocity) strongly related to length of delay of self

**Table 2 antibiotics-09-00285-t002:** Reviewed studies of psychological determinants of antibiotic (AB)/antimicrobial (AM) usage and AB/AM resistance (ABR)/(AMR) in farmed animals.

Authors, Year, Location	Study Type	Sample	Behaviors of Interest	Key Findings
Hockenhull et al., 2017 [27]; UK, Europe, US, South America, Africa	Rapid evidence assessment	48 studies of AM use in food-producing animals, including surveys (postal, telephone) and qualitative (interviews, focus groups) with farmers and veterinarians; 21 < n < 3004 participants	Attitudes, beliefs and external influences on AM usage	Drivers/barriers to AM reduction were related to practical factors (cost/animal health) rather the risks of AMR Most believed reducing AM use a good thing but insufficient evidence of AMR threat to human health Farmers did not always seek veterinarian advice when deciding whether/when to use ABs Concern that reducing AM usage would lead to unhealthy animals/lower production
Coyne et al., 2014 [28]; UK	Qualitative	6 focus groups: pig veterinarians and farmers; n = 26 participants	Drivers and motivations of AM use	Veterinarians influenced by external pressures (e.g., clients’/public’s views) Farmers’ concerns related to production and management of their farm
Coyne et al., 2019 [30]; UK	Mixed methods	Pig farmers: semi-structured qualitative interviews; n = 22 participants followed by questionnaire survey; n = 261 participants	Drivers to AM behaviors and attitudes towards responsible use	Reduced AM use associated with good management practices, low stocking densities and good animal health High cost of AMs could be a driver to reduce usage for farmers Vaccination viewed as a feasible alternative to AM use by farmers Few quoted concerns about AMR as a driver to reducing AM use
Golding et al., 2019 [31]; UK	Qualitative	Semi-structured interviews with cattle, sheep and pig veterinarians and farmers; n = 25 participants	Attitudes to own role in AMR resistance and stewardship	Veterinarians and farmers had good understanding of stewardship issues Treatment behavior did not always tally with beliefs Fear of negative health outcomes led to inappropriate usage of AMs Perceived poor practice in others undermined own efforts at stewardship
Friedman et al., 2007 [32]; US	Qualitative	Four focus groups with dairy farmers; n = 22 participants	Sources of information and knowledge about antibiotic resistance	Only 40% of farmers familiar with AMR 86% unconcerned that AB overuse could lead to resistance amongst farm workers Lack of finances and time were barriers to optimal use of AMs Advice from veterinarians and other farmers more important than scientific reports when making AM decisions
Ekakoro et al., 2019 [33]; US	Qualitative	Five focus groups with beef cattle producers; n = 39 participants	Drivers, alternatives, knowledge and perceptions for AM usage	Practical considerations (e.g., cost), influence of veterinarians/peers, and animal welfare determinants of AM usage Concerns over AMR not a driver Vaccination and good management practices viewed as valid alternatives
Kramer et al., 2017 [34]; Netherlands	Questionnaire-based survey	Dairy, veal and pig farmers; n = 457 participants	Motivating and enabling factors of AM usage	Four psychological factors identified: ‘referent beliefs’ (external non-vet sources of advice), ‘perceived risk’ (AMR risk perception), ‘knowledge’ (of infection routes and effectiveness of AMs), and ‘undesired attitude to regulations’ (disregard for AMR regulations) Better knowledge significantly linked to lower AM usage
Visschers et al., 2015 [35]; Belgium, France, Germany, Sweden	Questionnaire-based survey	Pig farmers; n = 215 participants	Motivators and barriers to AM usage	Perceived risk of AM usage related to actual usage of AMs Farmers perceived many benefits of AM usage but relatively few risks
Jones et al., 2015 [36]; UK	Questionnaire-based survey	Dairy farmers; n = 71 participants	Theory of Planned Behavior: attitudes and behavior regarding AB usage	Weak association between past behavior and intention to use AMs Intention strongly predicted by social norms, including perceived approval by veterinarians and other farmers More commercially aware farmers more likely to intend to reduce AB use
McIntosh & Dean, 2013 [37]; US	Literature review of US-based studies	Cattle feedlot industry; no details of reviewed studies given	Theory of Planned Behavior: social norms	Strong influence of social norms (including veterinarians, consumers, pharmaceutical companies, regulatory bodies) in decision-making re: AM use Source of influence differed depending on the animal’s health (e.g., acute vs. chronically sick
Jan et al., 2010 [38]; US	Mixed methods	Qualitative interviews with cattle feedlot industry veterinarians; n = 35 participants followed by questionnaire; n = 103 participants	Theory of Planned Behavior: social norms (client/farmer) applied to perceived risk of AMR	Pressure from others (including farmers and farm owners) to use ABs associated with lower perceived risk of ABs Degree of perceived risk varied with clinical indication (e.g., chronically versus acutely sick cattle)
Swinkels et al., 2015 [39]; Netherlands, Germany	Qualitative	Semi-structured interviews with dairy farmers; n = 39 participants	Social influences on decisions re: AM treatment	Strong effect for views of other farmers in influencing length of treatment Extended treatment being associated with the social norm of being ‘a good farmer’
Sadiq et al., 2018 [40]; Malaysia	Questionnaire-based survey	Ruminant farmers; n = 84 participants	Knowledge, attitudes and behavior towards AMR	Poor awareness of AMR and its impact on animals and public health Farmers believed sick animals need ABs and AMs do not have any side-effects Inappropriate usage evident: using AMs when not prescribed by a vet, storing AMs for later use.
Ström et al., 2018 [41]; Cambodia	Questionnaire-based survey	Interviews with person responsible for treating sick pigs on farm; n = 91 pig farms	Knowledge, attitudes and behavior towards AM usage	Nearly 50% had not heard of AMR Farmers relied on their own judgement regarding AB treatment and made their own decisions re: dosage and duration Nearly 60% believed they hadn’t received enough information on AM usage in animals
Coyne et al., 2019 [42]; Indonesia, Thailand, Vietnam	Literature review and case studies (mixed methods: interview and questionnaire)	Pigs, chickens and fish farmers: case studies n = 51 pig farms, n = 419 chicken farms	Drivers and behavior regarding AM usage	Drivers of higher AM usage were profitability and disease and mortality prevention Cost of AMs low compared to other production costs so farmers saw an economic advantage to AM use
Pham et al., 2015 [43]; Vietnam	Questionnaire-based survey	Interviews with fish farmers (freshwater and shrimp); n = 94 farms	Knowledge and opinions towards AM use	45% of farmers did not believe antibiotics were effective in treating disease, but 72% used ABs regularly Only 16% aware of regulations regarding AB use Farmers sought advice from drug manufacturers/sellers and not veterinarians Both using/not using AMs linked more to economic factors and not to concerns about AMR

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
