# Peer review of "Antimicrobial Resistance in Humans and Animals: Rapid Review of Psychological and Behavioral Determinants"

_antibiotics, 2020, doi:10.3390/antibiotics9060285_

Round 1
Reviewer 1 Report
Dear authors
Thank you for having an opportunity to review this paper. It is clear that a number of studies has been reviewed; however, it is not clear what the focus of the paper is.
- The use of theory is somewhat unclear; you start introducing some theories but without saying why they may be useful; also the summary is very descriptive and conclusions are not clear
- the results seem to be a mixture of studies using a theory and some focusing on geographical area
- it is also unclear why the focus of the paper is both human and animal related AMR; these are two very different areas;
- It is unclear why qualitative and quantitative findings are described together
I think this paper would benefit from clearer focus for each paragraph according to the theory etc; more justification or analysis on bringing quantitative and qualitative data together and perhaps stronger discussion: for example why the theory was useful; what are the key messages?
Reviewer 2 Report
The manuscript is interesting but is not in the aim /scope of the Journal that requires a “scientific approach” in the fields of biochemistry, chemistry, genetics, microbiology or pharmacology.
I would advise to submit this manuscript to other Journals that the Authors surely know better than me (also belonging to the MDPI group as for example European Journal of Investigation in Health, Psychology and Education or Behavioral Sciences) where it can attract more consideration from the readers.
Author Response
Reviewer's comments
The manuscript is interesting but is not in the aim /scope of the Journal that requires a “scientific approach” in the fields of biochemistry, chemistry, genetics, microbiology or pharmacology. I would advise to submit this manuscript to other Journals that the Authors surely know better than me (also belonging to the MDPI group as for example European Journal of Investigation in Health, Psychology and Education or Behavioral Sciences) where it can attract more consideration from the readers.
Response:
We believe that our paper does meet the scope of Antibiotics as detailed in our cover letter: "We believe that our paper is particularly appropriate for publication in Antibiotics which provides an international and interdisciplinary forum for the dissemination of research on all aspects of the uses of antibiotics including: their use in animals and in agriculture; antibiotic resistance and misuse; antimicrobial stewardship; and research exploring the determinants of antimicrobial use and resistance – all topics covered in our review." Further, we should point out that the field of psychology, as a scientific discipline, is subject to the same evidential rigour as other disciplines mentioned by the reviewer.
Reviewer 3 Report
In the submitted work, authors reviewed the current evidence examining psychological issues regarding the use of antibiotics and antimicrobials and resistance to these in both human and animal populations. The research topic is very interesting and the findings presented here are important to the field. Food producing animals are in general a reservoir of AMR, but less awareness of AMR in farmers/policymakers. I have several suggestions to help improve this work.
- I’d suggest authors summarize reviewed literatures/papers in a table, laying out the key findings/conclusions in a way to help readers grasp the point a lot easier (e.g. manifest differences of AMR awareness between humans and food animals).
- The body of this review mainly focused on reviewing published research articles, but the government definitely plays a significant role in controlling/mitigating AMR. Did the author dig into the national regulations of antibiotic use? For example, the USA banned the use of antibiotics in animal feed since 2017. A separate section describing the policy variances across countries would be another solid contribution to this field.
- Line 34: awkward wording, fix the sentence
Round 2
Reviewer 1 Report
Thank you for the revised manuscript. I have no further comments or suggestions and I think the paper has a clearer focus now.
Reviewer 2 Report
As I reported, I do not see any fields and scope from the official page of Antibiotics